# PMI-Masking:
# Principled masking of correlated spans

**Yoav Levine**      **Barak Lenz**      **Opher Lieber**      **Omri Abend**

**Kevin Leyton-Brown**      **Moshe Tennenholtz**      **Yoav Shoham**

AI21 Labs, Tel Aviv, Israel

{*yoavl,barakl,opherl,omria,...*}*@ai21.com*

## ABSTRACT

Masking tokens uniformly at random constitutes a common flaw in the pretraining of Masked Language Models (MLMs) such as BERT. We show that such uniform masking allows an MLM to minimize its training objective by latching onto shallow local signals, leading to pretraining inefficiency and suboptimal downstream performance. To address this flaw, we propose PMI-Masking, a principled masking strategy based on the concept of Pointwise Mutual Information (PMI), which jointly masks a token $n$-gram if it exhibits high collocation over the corpus. PMI-Masking motivates, unifies, and improves upon prior more heuristic approaches that attempt to address the drawback of random uniform token masking, such as whole-word masking, entity/phrase masking, and random-span masking. Specifically, we show experimentally that PMI-Masking reaches the performance of prior masking approaches in half the training time, and consistently improves performance at the end of training.

## 1 INTRODUCTION

In the couple of years since BERT was introduced in a seminal paper by Devlin et al. (2019a), Masked Language Models (MLMs) have rapidly advanced the NLP frontier (Sun et al., 2019; Liu et al., 2019; Joshi et al., 2020; Raffel et al., 2019). At the heart of the MLM approach is the task of predicting a masked subset of the text given the remaining, unmasked text. The text itself is broken up into tokens, each token consisting of a word or part of a word; thus "chair" constitutes a single token, but out-of-vocabulary words like "e-igen-val-ue" are broken up into several sub-word tokens. In BERT, 15% of tokens are chosen to be masked uniformly at random. It is the random choice of single tokens that we address in this paper: we show that this approach is suboptimal and offer a principled alternative.

To see why Random-Token Masking is suboptimal, consider the special case of sub-word tokens. Given the masked sentence "To approximate the matrix, we use the eigenvector corresponding to its largest e-[mask]-val-ue", an MLM will quickly learn to predict "igen" based only on the context "e-[mask]-val-ue", rendering the rest of the sentence redundant. The question is whether the network will also learn to relate the broader context to the tokens comprising "eigenvalue". When they are masked together, the network is forced to do so, but such masking occurs with vanishingly small probability. One might hypothesize that the network would nonetheless be able to piece such meaning together from local cues; however, we show that it often struggles to do so.

We establish this via a controlled experiment, in which we reduced the size of the vocabulary, thereby breaking more words into sub-word tokens. We compared the extent to which such vocabulary reduction degraded regular BERT relative to so-called Whole-Word Masking BERT (WWBERT) (Devlin et al., 2019b), a version of BERT that jointly masks all sub-word tokens comprising an out-of-vocabulary word during training. We show that vanilla BERT's performance degrades much more rapidly than that of WWBERT as the vocabulary size shrinks. The intuitive explanation

is that Random-Token Masking is wasteful; it overtrains on easy sub-word tasks (such as predicting "igen") and undertrains on harder whole-word tasks (predicting "eigenvalue").

The advantage of Whole-Word Masking over Random-Token Masking is relatively modest for standard vocabularies, because out-of-vocabulary words are rare. However, the tokenization of words is a very special case of a much broader statistical linguistic phenomenon of *collocation*: the co-occurrence of series of tokens at levels much greater than would be predicted simply by their individual frequencies in the corpus. There are millions of collocated word $n$-grams — multi-word expressions, phrases, and other common word combinations — whereas there are only tens of thousands of words in frequent use. So it is reasonable to hypothesize that Random-Token Masking generates many wastefully easy problems and too few usefully harder problems because of multi-word collocations, and that this affects performance even more than the rarer case of tokenized words; we show that this indeed is the case.

Several prior works have considered the idea of masking across spans longer than a single word. Sun et al. (2019) and Guu et al. (2020) proposed Knowledge Masking and Salient Span Masking, respectively, in which tokens comprising entities or phrases, as identified by external parsers, are jointly masked. While extending the scope of Whole-Word Masking, the restriction to specific types of correlated $n$-grams, along with the reliance on imperfect tools for their identification, has limited the gains achievable by these approaches. With a similar motivation in mind, SpanBERT of Joshi et al. (2020) introduced Random-Span Masking, which masks word spans of lengths sampled from a geometric distribution at random positions in the text. Random-Span Masking was shown to consistently outperform Knowledge Masking, is simple to implement, and inspired prominent MLMs (Raffel et al., 2019). However, while Random-Span Masking increases the chances of masking collocations, with high probability the selected spans break up correlated n-grams, such that the prediction task can often be performed by relying on local cues.

In this paper we offer a principled approach to masking spans that consistently provide high signal, unifying the intuitions behind the above approaches while also outperforming them. Our approach, dubbed *PMI-Masking*, uses Pointwise Mutual Information (PMI) to identify collocations, which we then mask jointly. At a high level, PMI-Masking consists of two stages. First, given any pretraining corpus, we identify a set of contiguous $n$-grams that exhibit high co-occurrence probability relative to the individual occurrence probabilities of their components. We for-

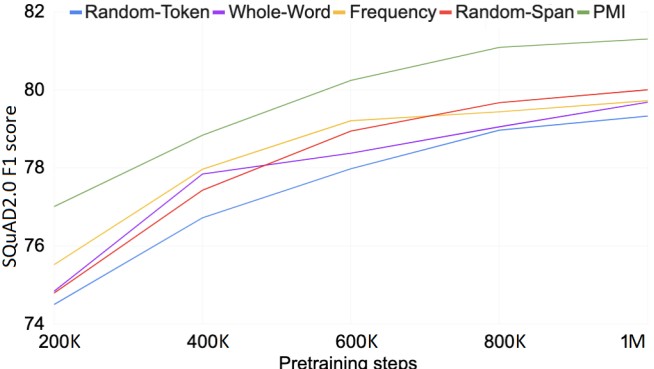

Figure 1: SQuAD2.0 development set F1 scores of BERT$_{\text{BASE}}$ models trained with different masking schemes, evaluated every 200K steps during pretraining.

malize this notion by proposing an extended definition of Pointwise Mutual Information from bigrams to longer $n$-grams. Second, we treat these collocated $n$-grams as single units; the masking strategy selects at random both from these units and from standard tokens that do not participate in such units. Figure 1, detailed and reinforced by further experiments in section 5, shows that (1) PMI-Masking dramatically accelerates training, matching the end-of-pretraining performance of existing approaches in roughly half of the training time; and (2) PMI-Masking improves upon previous masking approaches at the end of pretraining.

## 2 MOTIVATION: MLMS ARE SENSITIVE TO TOKENIZATION

In this section we describe a simple experiment that motivates our PMI-Masking approach. We examined BERT's ability to learn effective representations for words consisting of multiple sub-word tokens, treating this setting as an easily controlled analogue for the multi-word collocation problem that truly interests us. Our experiment sought to assess the performance gain obtained from always masking whole words as opposed to masking each individual token uniformly at random. We

|  | 1.08 **tokens per word**
(30K vocabulary) | 1.22 **tokens per word**
(10K vocabulary) | 2.06 **tokens per word**
(2K vocabulary) |
|---|---|---|---|
| Random-Token Masking | 79.3 | 77.8 | 72.8 |
| Whole-Word Masking | 79.7 | 79.5 | 77.6 |

Table 1: SQuAD2.0 development set F1 scores of BERT$_{\text{BASE}}$ models trained with Random-Token and Whole-Word masking schemes and with different vocabulary sizes (30K; 10K; 2K).

compared performance across a range of vocabulary sizes, using the same WordPiece Tokenizer[1] that produced the original vocabulary of $\sim$ 30K tokens. As we decreased a 30K-token vocabulary to 10K and 2K tokens, the average length of a word over the pretraining corpus increased from 1.08 tokens to 1.22 and 2.06 tokens, respectively. Thus, by reducing the vocabulary size, we increased the frequency of multi-token words by a large factor.

Table 1 presents the performance of BERT models trained with these vocabularies, measured as score on the SQuAD2.0 development set (the experimental setup is described in section 4). The downstream performance of Random-Token Masking substantially degraded as vocabulary size decreased and the number of spans of sub-word tokens increased. One reason for such degradation might be the model seeing less text as context (512 input tokens cover less text when more words are broken into multiple tokens). This possibly plays a role; however, for models with the same vocabularies trained via Whole-Word Masking, this degradation was significantly attenuated. We therefore conjecture that this degradation occurred primarily because of the random masking strategy, which allows the model to use "shortcuts" for minimizing its loss, thus hindering its ability to learn the distribution of the entire multi-token word.

If our conjecture is correct, such shortcuts are just as problematic in the case of inter-word collocations. In fact, for the regular 30K-token vocabulary, divided words are rare, so inter-word collocations would pose a larger problem than intra-word collocations in the common setting. One possible mitigation might be to expand the vocabulary to include multi-word collocations. However, there are millions of these, and such vocabulary sizes are currently infeasible. Even if we could get around the practical issue of size, this approach may suffer from generalization problems: the frequency of each multi-word collocation can be lower than the sample complexity for learning a meaningful representation. An alternative, more practical approach is to leave the vocabulary as is, but jointly mask co-located words, with the intention of cutting off local statistical "shortcuts" and allowing the model to improve further by learning from broader context. This is the approach we take in this paper. In what follows we detail such a masking approach and show its advantages experimentally.

## 3 MASKING CORRELATED $n$-GRAMS

### 3.1 EXISTING MASKING APPROACHES

We now more formally present the MLM setup as well as existing masking approaches, which we implement as baselines. Given text tokenized into a sequence of tokens, Masked Language Models are trained to predict a set fraction of "masked" tokens, where this fraction is called the masking budget and is traditionally set to 15%. The modified input is inserted into the Transformer-based architecture (Vaswani et al., 2017) of BERT, and the pretraining task is to predict the original identity of each chosen token. Several alternatives have been proposed for choosing the set of tokens to mask.

**Random-Token Masking (Devlin et al., 2019a)**  The original BERT implementation selects tokens for masking independently at random, where 80% of the 15% chosen tokens are replaced with [MASK], 10% are replaced with a random token, and 10% are kept unchanged.

**Whole-Word Masking (Devlin et al., 2019b)**  The sequence of input tokens is segmented into units corresponding to whole words. Tokens for masking are then chosen by sampling entire units at random until the masking budget is met. Following Devlin et al. (2019a), for 80%/10%/10% of the units, all tokens are replaced with [MASK] tokens/ random tokens/ the original tokens, respectively.

---

[1]https://github.com/huggingface/tokenizers

**Random-Span Masking (Joshi et al., 2020)** Contiguous random spans are selected iteratively until the 15% masking budget is spent. At each iteration, a span length (in words) is sampled from a geometric distribution $\ell \sim \text{Geo}(0.2)$, and capped at 10 words. Then, the starting point for the span to be masked is randomly selected. Replacement with `[MASK]`, random, or original tokens is done as above, where spans constitute the units.

## 3.2 PMI: FROM BIGRAMS TO $n$-GRAMS

Our aim is to define a masking strategy that targets correlated sequences of tokens in a principled way. Of course, modeling such correlations in large corpora was widely studied in computational linguistics (Zuidema (2006); Ramisch et al. (2012); *inter alia*). Particularly relevant to our work is the notion of Pointwise Mutual Information (Fano, 1961), which quantifies how often two events occur, compared with what we would expect if they were independent. Define the probability of any $n$-gram as the number of its occurrences in the corpus divided by the number of all the $n$-grams in the corpus. PMI leverages these probabilities to give a natural measure of collocation of bigrams: how surprising the bigram $w_1 w_2$ is, given the unigram probabilities of $w_1$ and $w_2$. Formally, given two tokens $w_1$ and $w_2$, the PMI of the bigram "$w_1 w_2$" is

$$\text{PMI}(w_1 w_2) = \log \frac{p(w_1 w_2)}{p(w_1)p(w_2)}. \tag{1}$$

Importantly, PMI is qualitatively different from pure frequency: a relatively frequent bigram may not have a very high PMI score, and vice versa. For example, the bigram "*book is*" appears $34772$ times in the WIKIPEDIA+BOOKCORPUS dataset but is ranked around position $760$K in the PMI ranking for bi-grams over this corpus, while the bigram "*boolean algebra*" appears $849$ times in the corpus but is ranked around position $16$K in the PMI ranking.

What about contiguous spans of more than two tokens? For a given $n$-gram, we would again like to measure how strongly its components indicate one another. We thus require a measure that captures correlations among more than two variables. A standard and direct extension of the PMI measure to more than two variables, referred to as 'specific correlation' in Van de Cruys (2011), and as 'Naive-$\text{PMI}_n$' in this paper, is based on the ratio between the $n$-gram's probability and the probabilities of its component unigrams:

$$\text{Naive-PMI}_n(w_1 \ldots w_n) = \log \frac{p(w_1 \ldots w_n)}{\prod_{j=1}^{n} p(w_j)} \tag{2}$$

As in the bivariate case, this measure compares the actual empirical probability of the $n$-gram in the corpus with the probability it would have if its components occurred independently. However, the above definition suffers from an inherent flaw: an $n$-gram's Naive-$\text{PMI}_n$ will be high if it contains a segment with high PMI, even if that segment is not particularly correlated with the rest of the $n$-gram. Consider for example the case of trigrams:

$$\text{Naive-PMI}_3(w_1 w_2 w_3) = \log\{\frac{p(w_1 w_2 w_3)}{p(w_1)p(w_2)p(w_3)} \cdot \frac{p(w_1 w_2)}{p(w_1 w_2)}\} = \text{PMI}(w_1 w_2) + \log \frac{p(w_1 w_2 w_3)}{p(w_1 w_2)p(w_3)}$$

Where $\text{PMI}(w_1 w_2)$ is defined in eq. 1. When $\text{PMI}(w_1 w_2)$ is high, the Naive-$\text{PMI}_3$ measure of the trigram "$w_1 w_2 w_3$" will start at this high baseline. The added term of $\log \frac{p(w_1 w_2 w_3)}{p(w_1 w_2)p(w_3)}$ quantifies the actual added information of "$w_3$" to this correlated bigram, *i.e.*, it quantifies how far $p(w_1 w_2 w_3)$ is from being separable w.r.t. the segmentation into "$w_1 w_2$" and "$w_3$". For example, since the PMI of the bigram "*Kuala Lumpur*" is very high, the Naive-$\text{PMI}_n$ of the trigram "*Kuala Lumpur is*" is misleadingly high, placing it at position $43$K out of all trigrams in the WIKIPEDIA+BOOKCORPUS dataset. It is in fact placed much higher than obvious collocations such as the trigram "*editor in chief*", which is ranked at position $210$K out of all trigrams.

In order to favor $n$-grams that cannot be easily subdivided into shorter unrelated spans, we propose a measure of distance from separability with respect to all of an $n$-gram's possible segmentations rather than with respect only to the segmentation into single tokens:

$$\text{PMI}_n(w_1 \ldots w_n) = \min_{\sigma \in \text{seg}(w_1 \ldots w_n)} \log \frac{p(w_1 \ldots w_n)}{\prod_{s \in \sigma} p(s)} \tag{3}$$

Here, seg($w_1 \ldots w_n$) is the set of all contiguous segmentations of the $n$-gram "$w_1 \ldots w_n$" (excluding the identity segmentation), where any segmentation $\sigma \in$ seg($w_1 \ldots w_n$) is composed of sub-spans which together give "$w_1 \ldots w_n$". Intuitively, this measure effectively discards the contribution of high PMI segments; the minimum in Eq. 3 implies that an $n$-gram's collocation score is given by its weakest link, *i.e.*, by the segmentation that is closest to separability. When ranked by the above $PMI_n$ measure, the trigram "*Kuala Lumpur is*" is demoted to position 1.6M, since the segmentation into "*Kuala Lumpur*" and "*is*" yields unrelated segments, while the trigram "*editor in chief*" is upgraded to position 33K since its segmentations yield correlated components. As we will see, this definition is not only conceptually cleaner, but also leads to improved performance.

### 3.2.1 PMI-Masking

We implement our strategy of treating highly collocating $n$-grams as units for masking by assembling a list of $n$-grams as a *masking vocabulary* in parallel to the 30K-token vocabulary. Specifically, we make use of the entire pretraining corpus for compiling a list of collocations. We consider word $n$-grams of lengths 2–5 having over 10 occurrences in the corpus, and include the highest ranking collocations over the corpus, as measured via our proposed $PMI_n$ measure (Eq. 3). Noticing that the $PMI_n$ measure is sensitive to the length of the $n$-gram, we assemble per-length rankings for each $n \in \{2, 3, 4, 5\}$, and integrate these rankings to compose the masking vocabulary. After conducting a preliminary evaluation of how an $n$-gram's quality as a collocation degrades with its $PMI_n$ rank (detailed in the appendix), we chose the masking vocabulary size to be 800K, for which approximately half of pretraining corpus tokens were identified as part of some correlated $n$-gram.

In order to get some sense of the differences between the attained masking vocabulary and prior approaches, we annotated a random sample of 500 bigrams and 500 trigrams from the masking vocabulary with Entity/Not-Entity labels. We found that only around 14% of the entries in the bigrams/trigrams lists were annotated as entities, while the rest are other types of collocations. Moreover, we found that while named entities are very prevalent at the very top of the list, they are scarce otherwise. By refining the view into highest and lowest ranking PMI bigram groups, we get that 50% of the top 20% are entities while only 1% of the bottom 20% are entities, and similar trends are attained for trigrams. This breakdown can illuminate a natural intuition regarding high ranking PMI n-grams representing entities (employed also by previous works (Downey et al., 2007; Korkontzelos et al., 2008)) – indeed the top ranking PMI entries are largely entities. But we chose a much larger PMI-based masking vocabulary (see appendix 1 on the process of choosing its size), and the proportion of entities drops to around 1/7, with many of the added entries representing other types of collocations (the annotated lists are given as supplementary material).

After composing the masking vocabulary, we treat its entries as units to be masked together. All input tokens not identified with entries from the masking vocabulary are treated independently as units for masking according to the Whole-Word Masking scheme. If one masking vocabulary entry contains another entry in a given input, we treat the larger one as the unit for masking, *e.g.*, if the masking vocabulary contains the $n$-grams "the united states", "air force", and "the united states air force", the latter will be one unit for masking when it appears. In the case of overlapping entries, we choose one at random as a unit for masking and treat the remaining tokens as independent units, *e.g.*, if the input text contains "by the way out" and the masking vocabulary contains the $n$-grams "by the way" and "the way out", we can choose either "by the way" and "out" or "by" and "the way out" as units for masking.

After we segment the sequence of input tokens into units for masking, we then choose tokens for masking by sampling units uniformly at random until 15% of the tokens (the standard tokens of the 30K-token vocabulary) in the input are selected. As in the prior methods, replacement with [MASK] (80%), random (10%), or original (10%) tokens is done at the unit level.

## 4 Experimental setup

To evaluate the impact of PMI-Masking, we trained Base-sized BERT models (Devlin et al., 2019a) with each of the masking schemes presented in Section 3. Rather than relying on existing implementations for baseline masking schemes, which vary in training specifics, we reimplemented each scheme within the same framework used to train our PMI-Masked models. For control, we trained within the same framework models with Naive-PMI-Masking and Frequency-Masking, following

the procedure described above for PMI-Masking, but ranking by the Naive-PMI$_n$ measure (Eq. 2) and by pure-frequency, respectively. In Section 5, we compare our PMI-Masking to all internally-trained masking schemes (Table 2) as well as with externally released models (Table 3).

## 4.1 PRETRAINING

We trained uncased models with a 30K-sized vocabulary that we constructed over WIKIPEDIA +BOOKCORPUS via the WordPiece Tokenizer used in BERT. We omitted the Next Sentence Prediction task, as it was shown to be superfluous (Joshi et al., 2020), and trained only on the Masked Language Model task during pretraining. We trained with a sequence length of 512 tokens, batch size of 256, and a varying number of steps detailed in Section 5. For pretraining, after a warmup of 10,000 steps we used a linear learning rate decay, therefore models that ran for a different over-all amount of steps are not precisely comparable after a given amount of steps. We set remaining parameters to values similar to those used in the original BERT pretraining, detailed in the appendix.

We performed the baseline pretraining over the original corpus used to train BERT: the 16GB WIKIPEDIA+BOOKCORPUS dataset. We show that PMI-Masking achieved even larger performance gains relative to the baselines when training over more data, by adding the 38GB OPEN-WEBTEXT (Gokaslan & Cohen, 2019) dataset, an open-source recreation of the WebText corpus described in Radford et al. (2019). As described in section 3, we compose our PMI$_n$-based masking vocabulary according to the pretraining corpus in use.

## 4.2 EVALUATION

We evaluate our pretrained models on two question answering benchmarks: the Stanford Question Answering Dataset (SQuAD) and the ReAding Comprehension from Examinations (RACE), as well as on the General Language Understanding Evaluation (GLUE) benchmark. Additionally, we report the Single-Token perplexity of our pretrained models.

- **SQuAD** (Rajpurkar et al., 2016) has served as a major question answering benchmark for pre-trained models. It provides a paragraph of context and a question, and the task is to answer the question by extracting the relevant span from the context. We focus on the latest more challenging variant, SQuAD2.0 (Rajpurkar et al., 2018), in which some questions are not answered in the provided context, and the task includes identifying such cases.

- **RACE** (Lai et al., 2017) is a large-scale reading comprehension dataset collected from English examinations in China, designed for middle and high school students. Each passage is associated with multiple questions; for each, the task is to select one correct answer from four options. RACE has significantly longer context than other popular reading comprehension datasets and the proportion of questions that requires reasoning is very large.

- **GLUE** (Wang et al., 2018) is a collection of 9 datasets for evaluating natural language understanding systems. Tasks are framed as either single-sentence classification or sentence-pair classification tasks. For full details, please see the appendix.

- **Single-Token perplexity** We evaluate an MLM's ability to predict single-tokens by measuring perplexity over a held out test set of 110K tokens from OPENWEBTEXT. For each test example, a single token for prediction is masked and the remainder of the input tokens are unmasked.

In Tables 2 and 3, for every downstream task we swept 8 different hyperparameter configurations (batch sizes $\in \{16, 32\}$ and learning rates $\in \{1, 2, 3, 5\} \cdot 10^{-5}$). We report the best median development set score over five random initializations per hyper-parameter. When applicable, the model with this score was evaluated on the test set. The development set score of each configuration was attained by fine-tuning the model over 4 epochs (SQuAD2.0 and RACE) or 3 epochs (all GLUE tasks except RTE and STS – 10 epochs) and performing early stopping based on each task's evaluation metric on the development set. In the preliminary experiments of Table 1, and in Figures 1 and 2 for which we evaluate many pretraining checkpoints per model, we report the average of the three middle scores out of 5 random initializations for a single set of hyper-parameters (batch size 32 and learning rate $3 \cdot 10^{-5}$).

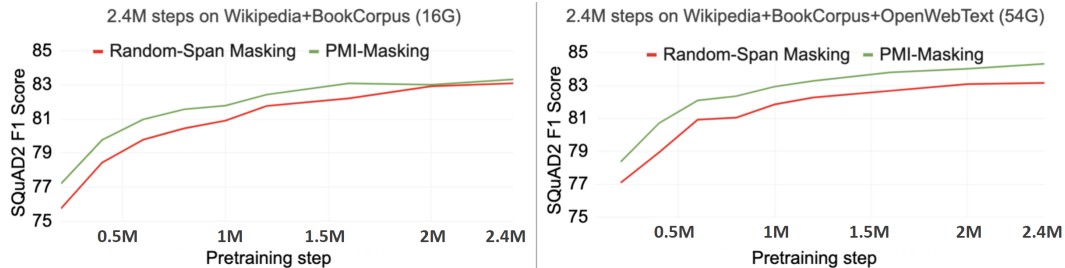

Figure 2: Scores on SQuAD2.0 development set of $BERT_{BASE}$ models trained for 2.4M steps, as done by Joshi et al. (2020) when proposing Random-Span Masking. Left: PMI-Masking efficiently elicits information from limited data. Right: More data, PMI-Masking continues to improve. See numerical scores in the appendix, along with the same trends on the RACE benchmark.

## 5 EXPERIMENTAL RESULTS

We evaluated the different masking strategies in two key ways. First, we measured their effect on downstream performance throughout pretraining to assess how efficiently they used the pretraining phase. Second, we more exhaustively evaluated downstream performance of different approaches at the end of pretraining. We examine how the advantage of PMI-Masking is affected by the size of the pretraining corpus and by amount of examples seen during pretraining (batch size × training steps).

### 5.1 EVALUATING DOWNSTREAM PERFORMANCE THROUGHOUT PRETRAINING

By examining the model's downstream performance after each 200K steps of pretraining, we demonstrate that PMI-Masking speeds up MLM training. Figure 1 investigates the standard BERT setting of pretraining on the Wikipedia+BookCorpus dataset for 1M training steps with batch size 256. It shows that the PMI-Masking method clearly outperformed a variety of prior approaches, as well as the baseline pure frequency based masking, on the SQuAD2.0 development set for all examined checkpoints (these patterns are consistent on RACE, see detailed scores in the appendix). PMI-Masking achieved the score of Random-Span Masking, the best of the existing approaches, after roughly half as many steps of pretraining.

We ran a second experiment that increased the number of steps from 1M to 2.4M, while maintaining the batch size and the pretraining corpus; this was the setting used by Joshi et al. (2020) when proposing Random-Span Masking. We observed that while PMI-masking learned much more quickly, it eventually reached a plateau, and Random-Span Masking caught up after enough training steps. Figure 2 (left) details these results.

Finally, we increased the amount of training data by adding the OPENWEBTEXT corpus ($\sim 3.5\times$ more data). Figure 2 (right) demonstrates that the plateau we previously observed in PMI-Masking's performance was due to limited training data. When training for 2.4M training steps on the Wikipedia+BookCorpus+OpenWebText dataset, PMI-masking reached the same score that Random-Span Masking did at the end of training after roughly half of the pretraining, and continued to improve. Thus, PMI-Masking definitively outperformed Random-Span masking in a scenario where data was not a bottleneck, as is ideally the case in MLM pretraining (Raffel et al., 2019).

### 5.2 EVALUATING DOWNSTREAM PERFORMANCE AFTER PRETRAINING

Table 2 shows that after pretraining was complete, PMI-Masking outperformed prior masking approaches in downstream performance on the SQuAD2.0, RACE, and GLUE benchmarks. In agreement with Figure 2, for longer pretraining (2.4M training steps) the absolute advantage of PMI-Masking is boosted across all tasks when pretraining over a larger corpus (adding OPENWEBTEXT). The table also shows that Naive-PMI Masking, based on the straightforward extension in eq. 2 to the standard bivariate PMI, significantly falls behind our more nuanced definition in eq. 3, and is often on par with Random-Span Masking.

| BERT Base with different maskings | SQuAD2.0 EM | F1 | RACE Acc. | GLUE Avg |
|---|---|---|---|---|
| *1M training steps on* WIKIPEDIA+BOOKCORPUS*(16G):* | | | | |
| Random-Token Masking | 76.4/– | 79.6/– | 67.8/66.2 | 83.1/– |
| Random-Span Masking | 77.1/– | 80.3/– | 68.6/66.9 | 83/– |
| Naive-PMI-Masking | 78.2/– | **81.3**/– | 69.7/67.8 | **84.1**/– |
| PMI-Masking | **78.5**/– | **81.4**/– | **70.1/68.4** | **84.1**/– |
| *2.4M training steps on* WIKIPEDIA+BOOKCORPUS*(16G)* | | | | |
| Random-Span Masking | 79.7/80.0 | 82.7/82.8 | 71.9/69.5 | **84.8**/79.7 |
| Naive-PMI-Masking | **80.3**/80.2 | **83.2**/83.2 | 71.7/69.8 | 84.5/80.0 |
| PMI-Masking | 80.2/**80.9** | 83.3/ **83.6** | **72.3/70.9** | 84.7/**80.3** |
| *2.4M training steps on* WIKIPEDIA+BOOKCORPUS+OPENWEBTEXT*(54G):* | | | | |
| Random-Span Masking | 80.1/80.4 | 83.2/83.3 | 74.0/72.2 | 85.1/80.1 |
| Naive-PMI-Masking | 80.4/80.0 | 83.3/83.0 | 73.9/71.4 | 85.6/80.3 |
| PMI-Masking | **80.9/82.0** | **83.9/84.9** | **74.8/73.2** | **86.0/80.8** |

Table 2: Dev/Test performance on the SQuAD, RACE, and GLUE benchmarks of BERT Base sized models pretrained and evaluated according to section 4. We report EM (exact match) and F1 scores for SQuAD2 and accuracy for RACE. For GLUE we report the average scores on the development set and the official leaderboard scores on the test set (see the per-task scores in the appendix).

| PMI vs Prior BASE MLMs | Corpus size | Batch $\times$ Steps = Examples | RACE dev/test |
|---|---|---|---|
| *PMI vs n-grams in vocabulary* | | | |
| AMBERT (Zhang & Li, 2020) | 47G | $1024 \times 0.5M = 512G$ | 68.9[†]/66.8[†] |
| PMI-Masking | 16G | $256 \times 1M = 256M$ | **70.1/68.4** |
| *PMI vs Random-Span Masking* | | | |
| SpanBERT$_{BASE}$ (Joshi et al., 2020) | 16G | $256 \times 2.4M = 614.4M$ | 70.5/68.7 |
| PMI-Masking | 16G | $256 \times 2.4M = 614.4M$ | **72.3/70.9** |
| *PMI vs Random-Token Masking with 3X more data and 6X more training examples* | | | |
| RoBERTa$_{BASE}$ (Liu et al., 2019) | 160G | $8K \times 0.5M = 4G$ | **74.9**/73 |
| PMI-Masking | 54G | $256 \times 2.4M = 614.4M$ | **74.8/73.2** |

Table 3: Comparing the RACE scores of our PMI-Masked models with comparable published Base-sized models. The scores of prior MLMs were attained by finetuning released models in the same setup of the PMI-Masked models (Section 4), except for those marked in '†', reported in Zhang & Li (2020). The number of examples reflects the amounts of text examined during training, as all prior models train over the same sequence length as our PMI-Masked models, namely 512. AMBERT was trained over WIKIPEDIA+OPENWEBTEXT (47G), SpanBERT over WIKIPEDIA+BOOKCORPUS (16G), and RoBERTa over WIKIPEDIA+BOOKCORPUS+OPENWEBTEXT+STORIES+CCNEWS (160G – see details in Liu et al. (2019)).

We also compared our PMI-Masking Base-sized models to published Base-sized models (Table 3), and again saw PMI-Masking increase both pretraining efficiency and end-of-training downstream performance. Zhang & Li (2020) trained their 'AMBERT' model over a vocabulary of $n$-grams in parallel to the regular word/subword level vocabulary, performing the hard task of $n$-gram prediction in parallel to the easy Random-Token level prediction task during pretraining. This approach yielded a model with 75% more parameters than the common Base size of our PMI-Masking model. By using the PMI-masking scheme on a regular BERT architecture and vocabulary, we attained a significantly higher score on the RACE benchmark, despite training over a corpus $3\times$ smaller and showing the model $2\times$ fewer examples during pretraining.

Joshi et al. (2020) and Liu et al. (2019) only reported scores for SpanBERT and RoBERTa (respectively) for Large-sized models in their original papers, but did release weights for Base-sized models. We fine-tuned these models on the RACE development set via the same fine-tuning procedure we employed for our PMI-Masking models (described in Section 4), and evaluated the best performing model on the publicly available RACE test set. A PMI-Masking Base-sized model scored more than 2 points higher than the SpanBERT$_{BASE}$ trained by Random-Span Masking over the same pretraining corpus when shown the same number of examples. Remarkably, a PMI-Masking Base-sized model scored slightly higher than RoBERTa$_{BASE}$ trained by Random-Token Masking, even though RoBERTa was given access to a pretraining corpus $3\times$ larger and shown $6\times$ more training examples.

Lastly, we note that the measure of Single-Token perplexity is not indicative of downstream performance, when reported for models trained with different masking schemes. Comparing the adjacent table with the downstream evaluation of the same models in Table 2, it is clear that the ability to predict single tokens from context is not correlated with performance. This reinforces our observation that by minimizing their training objective, standard MLMs, which mask tokens randomly, train to excel on relatively many easy tasks that do not reflect the knowledge required for downstream understanding.

| Single-Token Perplexity | |
| --- | --- |
| Random-Token Masking | **2.96** |
| Random-Span Masking | 4.30 |
| Naive-PMI-Masking | 7.35 |
| PMI-Masking | 21.85 |

Table 4: The Single-Token perplexity of MLMs trained for 1M steps over WIKI+BOOKCORPUS.

## 6 CONCLUSION

Bidirectional language models hold the potential to unlock greater signal from the training data than unidirectional models (such as GPT). BERT-based MLMs are historically the first (and still the most prominent) implementation of inherently bidirectional language models, but they come at a price. A hint of this price is the fact that Single-Token perplexity, which captures the ability to predict single tokens and which has a natural probabilistic interpretation in the autoregressive unidirectional case, ceases to correlate with downstream performance across different MLMs (see Table 4). This means that the original MLM task, which is focused on single token prediction, should be reconsidered. This has been the focus of this paper, which points to the inefficiency of random-token masking, and offers PMI-masking as an alternative with several advantages: (i) It is a principled approach, based on a nuanced extension of binary PMI to the n-ary case. (ii) It leads to better downstream performance, for example it surpasses RoBERTa (which uses vanilla random token masking) on the challenging reading comprehension RACE test with $6\times$ less training over a $3\times$ smaller corpus, and it dominates the more naive, heuristic approach of random span masking at any point during pretraining, matches its end-of-training performance halfway during its own pretraining, and at the end of training improves on it by 1-2 points across a variety of downstream tasks. Perhaps due to their conceptual simplicity, unidirectional models were the first to break the 100B parameter limit with the recent GPT3 (Brown et al., 2020). Bidirectional models will soon follow, and this paper can accelerate their development by offering a way to significantly lower their training costs while boosting performance.

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

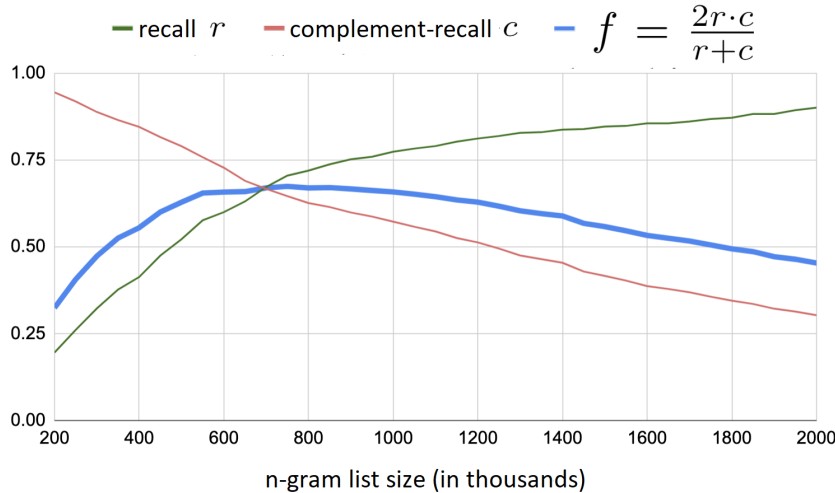

Figure 3: Quality measures of top ranking $\text{PMI}_n$ $n$-grams lists increased in increments of 50K. The masking vocabulary size was chosen such that it includes as many $n$-grams labeled as *collocation* as possible, while not including too many $n$-grams labeled as *not a collocation*, in an internally constructed test set detailed below. $r$ is the percent of all positively labeled examples from the test set that appear within the given list (recall), $c$ is the percent of all negatively labeled examples from the test set that *do not* appear within the given list (complement-recall). We aim for a list size for which both $r$ and $c$ are high enough, and employ $f$ as a measure for this, finally choosing a list size of 800K.

## A  DETERMINING THE MASKING VOCABULARY SIZE

The $\text{PMI}_n$ measure, defined in eq. 3, provides an $n$-gram ranking function that is intended to rank an $n$-gram higher if its components are more indicative of one another. However, this measure alone is not enough for composing a masking vocabulary: we need to decide on its size $M$ (the masking vocabulary will be composed of the top-$M$ ranked $n$-grams). One could advocate for an ablation study in which $M$ is varied, and models are pretrained per $M$ and evaluated. This can be done in future work, and perhaps an even stronger result can be shown for PMI-Masking with masking vocabulary size chosen by such optimization.

As a proxy, we determined the masking vocabulary size $M$ via a small scale evaluation of an $n$-gram's "collocation quality" as a function of its $\text{PMI}_n$ rank. Specifically, we created an ad hoc test set composed of 1000 $n$-grams that we labeled either as *collocation* or *not a collocation* (available upon request). We did that by choosing at random 10 words with frequency above 10000 in WIKIPEDIA+BOOKCORPUS, and for each word sampled 25 $n$-grams per length $n \in \{2, 3, 4, 5\}$ that contain it. Finally, we manually labeled each collected $n$-gram, where the textbook definition of collocation was given to the annotators (the annotator agreement was 80% over 100 shared examples).

Then, we increased a list size $M$ in steps of 50K, adding $n$-grams from the top ranking $\text{PMI}_n$ downwards. For each $M$-sized list we computed two different scores on the test set. The first is the recall of the positive examples in the list, denoted $r$: the percent of all positively labeled examples from the test set that appear within the given list. The second is the recall of the negative examples in the complement of the list, dubbed complement-recall, denoted $c$: the percent of all negatively labeled examples from the test set that *do not* appear within the given list. By these definitions, the recall $r$ starts low and increases with list size and the complement-recall $c$ follows an opposite trend, as can be seen in Figure 3. Our desired masking vocabulary size should yield a list with many $n$-grams labeled as *collocation* while containing little $n$-grams labeled *not a collocation*. we define $f = \frac{2r \cdot c}{r+c}$ as a measure for optimization which balances the two requirements, and Figure 3 shows that this measure is highest at sizes of around 700-800, so we set the masking vocabulary size to be 800K.

## B  PRETRAINING AND MODEL DETAILS

Table 5 shows the pretraining hyper-parameters we used, as well as the architecture specifics, both follow the standard implementation of BERT.

| | |
|---|---|
| Number of Layers | 12 |
| Hidden Size | 768 |
| Sequence Length | 512 |
| FFN Inner Hidden Size | 3072 |
| Attention Heads | 12 |
| Attention Head Size | 64 |
| Dropout | 0.1 |
| Attention Dropout | 0.1 |
| Warmup Steps | 10,000 |
| Peak Learning Rate | 1e-4 |
| Batch Size | 256 |
| Weight Decay | 0.01 |
| Initializer Range | 0.02 |
| Learning Rate Decay | Linear |
| Adam $\epsilon$ | 1e-6 |
| Adam $\beta_1$ | 0.9 |
| Adam $\beta_2$ | 0.999 |

Table 5: Hyper-parameters of the architecture and pretraining, complementing the description in Section 4.

## C  EVALUATION OF DIFFERENT CHECKPOINTS DURING PRETRAINING

Tables 6 and 7 respectively present the development set scores on SQuAD2.0 and RACE, attained for models at different checkpoints during pretraining. The SQuAD2.0 scores are depicted in Figures 1 and 2.

| pretraining checkpoint: | 200 | 400 | 600 | 800 | 1000 | 1200 | 1600 | 2000 | 2400 |
|---|---|---|---|---|---|---|---|---|---|
| *1M training steps on* WIKIPEDIA+BOOKCORPUS | | | | | | | | | |
| Random-Token Masking | 74.4 | 76.7 | 77.9 | 78.9 | 79.3 | – | – | – | – |
| Whole-Word Masking | 74.8 | 77.9 | 78.4 | 79.1 | 79.6 | | | | |
| Frequency-Masking | 75.5 | 78 | 79.2 | 79.4 | 79.7 | – | – | – | – |
| Random-Span Masking | 74.8 | 77.4 | 78.9 | 79.6 | 80.0 | – | – | – | – |
| PMI-Masking | **77.0** | **78.8** | **80.3** | **81.1** | **81.3** | – | – | – | – |
| *2.4M training steps on* WIKIPEDIA+BOOKCORPUS | | | | | | | | | |
| Random-Span Masking | 75.8 | 78.4 | 79.8 | 80.4 | 80.9 | 81.8 | 82.2 | **82.9** | 83.1 |
| PMI-Masking | **77.2** | **79.8** | **81.0** | **81.6** | **81.8** | **82.4** | **83.1** | 83.0 | **83.3** |
| *2.4M training steps on* WIKIPEDIA+BOOKCORPUS+OPENWEBTEXT | | | | | | | | | |
| Random-Span Masking | 77.1 | 78.9 | 80.9 | 81.0 | 81.8 | 82.3 | 82.7 | 83.1 | 83.2 |
| PMI-Masking | **78.4** | **80.7** | **82.1** | **82.4** | **82.9** | **83.3** | **83.8** | **84.0** | **84.3** |

Table 6: The F1 score on the SQuAD2.0 development set of models taken at various checkpoints along the pretraining of BERT Base sized models trained with different masking schemes. These scores are depicted in Figures 1 and 2. We finetuned on SQuAD2.0 with batch size of 32 and learning rate of $3 \cdot 10^{-5}$ over 4 epochs without early stopping. We did this for 5 random initializations of the task's head and the reported score is an average of the three middle scores.

| pretraining checkpoint: | 200 | 400 | 600 | 800 | 1000 | 1200 | 1600 | 2000 | 2400 |
|---|---|---|---|---|---|---|---|---|---|
| *1M training steps on* WIKIPEDIA+BOOKCORPUS | | | | | | | | | |
| Random-Token Masking | 61.2 | 64.3 | 65.6 | 66.4 | 67.1 | – | – | – | – |
| Whole-Word Masking | 62.0 | 64.9 | 66.0 | 67.0 | 67.8 | | | | |
| Frequency-Masking | 63.7 | 65.7 | 67.3 | 68.5 | 68.8 | – | – | – | – |
| Random-Span Masking | 61.7 | 64.7 | 66.8 | 67.9 | 68.0 | – | – | – | – |
| PMI-Masking | **63.5** | **66.8** | **68.4** | **68.9** | **69.7** | – | – | – | – |
| *2.4M training steps on* WIKIPEDIA+BOOKCORPUS | | | | | | | | | |
| Random-Span Masking | 62.3 | 64.3 | 65.6 | 67.8 | 69.0 | 68.9 | **70.3** | **71.0** | 71.4 |
| PMI-Masking | **63.6** | **66.7** | **67.3** | **68.5** | **69.2** | **70.4** | 70.5 | 71.2 | **72.2** |
| *2.4M training steps on* WIKIPEDIA+BOOKCORPUS+OPENWEBTEXT | | | | | | | | | |
| Random-Span Masking | 64.6 | 67.0 | 69.2 | 69.9 | 70.5 | 71.3 | 72.9 | 73.5 | 73.4 |
| PMI-Masking | **66.5** | **68.6** | **70.7** | **71.4** | **72.4** | **72.5** | **73.6** | **74.1** | **74.5** |

Table 7: The accuracy score on the RACE development set of models taken at various checkpoints along the pretraining of BERT Base sized models trained with different masking schemes. We finetuned on RACE with batch size of 32 and learning rate of $3 \cdot 10^{-5}$ over 4 epochs without early stopping. We did this for 5 random initializations of the task's head and the reported score is an average of the three middle scores.

## D  GLUE TASKS AND DETAILED SCORES

The General Language Understanding Evaluation (GLUE) benchmark (Wang et al., 2019) consists of 9 sentence-level tasks. Sentence-level classification tasks: CoLA (Warstadt et al., 2018) (evaluating linguistic acceptability) and SST-2 (Socher et al., 2013) (sentiment classification). Sentence-pair similarity tasks: MRPC (Dolan & Brockett, 2005) (binary paraphrasing classification task), STS-B (Cer et al., 2017): (graded similarity scoring task), and QQP[2] (binary paraphrasing classification task). Natural language inference tasks: MNLI (Williams et al., 2018), QNLI (Rajpurkar et al., 2016), RTE (Dagan et al., 2005; Bar-Haim et al., 2006; Giampiccolo et al., 2007) and WNLI (Levesque et al., 2011). Table 8 shows the detailed per-task scores of our examined models.

| GLUE | MNLI | QNLI | QQP | RTE | SST | MRPC | CoLA | STS | Avg |
|---|---|---|---|---|---|---|---|---|---|
| *1M training steps on Wikipedia+BookCorpus; on dev* | | | | | | | | | |
| Random-Span Masking | 84.0/- | 91.4 | 90.8 | 69.0 | 92.8 | 88.5 | 58.5 | 88.9 | 83.0 |
| Naive-PMI-Masking | **85.1**/– | **91.9** | **91.0** | **74.0** | **93.3** | 88.2 | 60.3 | **89.3** | **84.1** |
| PMI-Masking | **85.2**/– | 91.8 | **91.0** | 72.2 | 92.7 | **89.7** | 60.6 | **89.3** | **84.1** |
| *2.4M training steps on* WIKIPEDIA+BOOKCORPUS; *on test* | | | | | | | | | |
| Random-Span Masking | **85.7**/84.7 | **92.9** | **89.4** | **69.8** | 93 | 85.4 | 56.5 | 86.6 | 79.7 |
| Naive-PMI-Masking | 85.5/**85.3** | 92.2 | 89.2 | 68.9 | 93.6 | 85.4 | 59.4 | **87.3** | 80.0 |
| PMI-Masking | 85.3/85.0 | 92.0 | 89.2 | 69.0 | **94.0** | **85.6** | 61.8 | 86.8 | **80.3** |
| *2.4M training steps on* WIKIPEDIA+BOOKCORPUS+OPENWEBTEXT; *on test* | | | | | | | | | |
| Random-Span Masking | 86.3/85.1 | 92.2 | **89.4** | 71.1 | 94.6 | 85.6 | 56.8 | 87.2 | 80.1 |
| Naive-PMI-Masking | 86/85.4 | 91.7 | **89.4** | 69.2 | **95.1** | 87.8 | 57.5 | **87.9** | 80.3 |
| PMI-Masking | **86.6**/**85.8** | **93.1** | **89.5** | **72.9** | 94.7 | 87.7 | 57.4 | 87.7 | **80.8** |

Table 8: Results on the different tasks of the GLUE benchmark. For all tasks the scores reflect accuracy, except for STS-B (spearman score) and CoLA (Mathews Correlation). For results reported on the development set (1M training steps), the average score is simply the average of reported scores. For results reported on the test sets (2.4M training steps), the average score is the official GLUE leaderboard score. The official score includes averaging of F1 scores for QQP and MRPC, as well as the default majority submission score of 65.1 for WNLI.

---

[2]https://data.quora.com/First-Quora-Dataset-Release-Question-Pairs

