# OpenReview forum: "PMI-Masking: Principled masking of correlated spans"
_ICLR.cc/2021/Conference — ICLR 2021 Spotlight_

### Official Review · AnonReviewer1 · 2020-10-28
**Simple and elegant approach to improving MLM showing good results.**

**Rating:** 8
**Confidence:** 4

**Review:**

### Summary

This paper proposes an improvement to how tokens are selected for masking in pre-training large masked language models (BERT and family). Specifically, it stipulates that purely random choice of words (or word pieces) makes the MLM task insufficiently hard. It then goes on to propose a data-driven approach for selecting n-grams to mask together. The approach, based on an extension of pointwise mutual information for n-grams, is shown to outperform random token and random spans masking strategies on performance of downstream tasks.

### Strong and weak points

Strong points:
- The paper is very well written throughout, and easy to follow.
- The problem is well motivated with empirical evidence. I think Section 2 demonstrates well the case for random masking being too easy.
- The multivariate version of PMI proposed is simple and well motivated.
- The evaluation experiments are convincing the results are robust across the tasks shown,

Weak points:
 - The one main drawback in this study is the lack of comparison with entity-based techniques for masking.  In particular [1] has recently defined “salient span masking” based on named entity recognition and dates. Salient span masking has been adopted in [2] where it is shown to boost performance of open-domain question answering by 9+ points (Table1 of [2], models tagged with “+ SSM”).
I think it would be extremely interesting to compare these techniques (SSM specifically, but entity-based techniques in general) with PMI-Masking. Specifically, it is currently unclear whether the PMI based n-gram masking vocabulary simply ends up rediscovering popular named entity mentions, or whether there are more interesting sub-phrases (e.g., idiomatic sub-phrases) that a NER system would not select. Finally, it would be interesting to empirically test whether these extra non-entity n-grams provide further performance boost over the entity-based “salient span masking”.

[1] REALM: Retrieval-Augmented Language Model Pre-Training (https://arxiv.org/abs/2002.08909).
[2] How Much Knowledge Can You Pack Into the Parameters of a Language Model? (https://arxiv.org/abs/2002.08910)

### Recommendation

I recommend this paper for acceptance. The analysis and ideas throughout the paper are well executed. I also think the topic should be of high interest to the ICLR and NLP communities, given the importance of MLM pre-training on most state-of-the-art models at the moment. Despite the lack of comparison with entity-based techniques, having a statistically principled alternative, solely based on co-occurrence, without linguistic grounding, seems interesting.

### Questions for authors

1. The main question here related to the entity-based approaches discussed above. I think it would be interesting to address this issue given how closely related it is to this work, and the good performance the cited papers demonstrate using it. I can think of a couple of ways to address this comparison: (1) qualitative analysis of the masking vocabulary to better understand the differences between “PMI ngrams” and entity mentions, (2) experimental analysis incorporating some entity-based masking into the experiments in the paper.

2. Irrespectively of how you choose to address the entity-based comparison, I was interested in some analysis, or sampling, of the PMI-Masking vocabulary to understand what type of n-grams are being selected (entity mentions, idiomatic phrases, noun-phrases, etc.). Would be interesting if you could make this vocab available or add a small sample in the appendix.

3. Throughout the paper there is an assumption that contiguous words are considered for masking. This was not immediately clear in the beginning of the paper (I realized it only in Section 3.2 with - “What about contiguous spans of more than two tokens?”).  But one question came to mind: what about correlated non-contiguous spans? For example “eigenvalue” and “eigenvector” are unlikely to be present in the same n-gram, but have reasonably high chances of showing up together in the same passage. Have you considered extending this work to non-contiguous spans? Is there any expectation that this would help learning, or is it just a bad idea?

4. Was there an attempt to mix masking strategies during pre-training? Although Table 6 is convincing in demonstrating that single-token perplexity is not correlated with performance of downstream tasks, the differences seem curious. One is left wondering if there is any benefit in adding a small number of “easy” masking cases (i.e., random-tokens or random-spans)?

---

> ### Author Response · Authors · 2020-11-19
> **Thank you! Some stats showing PMI-Masking is different from entity-masking; Mixing-masking strategies is interesting to pursue**
>
> Thank you for your detailed feedback and support.
>
> Q1&2: Thank you for pointing out the SSM technique. While it resembles the idea of Ref [1] and the implementation in Ref [2] who showed random span masking to be better (and not reliant on parsers, like PMI-Masking), it is definitely worth mentioning. The success of SSM on open domain question answering encourages examination of PMI-Masking on such tasks. Our decision not to include entity based masking in our ablations relied mainly on Table 6 in Ref [2], who showed rather conclusive evidence for entity-masking underperforming relatively to their proposed Random-Span masking. Since PMI-Masking clearly improved upon Random-Span masking (the best existing masking strategy we are aware of), we decided to mainly compare to Random-Span masking.
> However, we do see why related questions can arise, and in order for the difference between our method and entity masking to be very clear, we will follow both your suggestions.
>
> (1) qualitative analysis of the masking vocabulary to better understand the differences between “PMI ngrams” and entity mentions:
> We have included in the supplementary material 2 files containing a random sample of 500 bigrams and trigrams from the list (one file each) with our annotation of Entity / Not Entity, which reveal an interesting picture. First, Only around 14/16% of the entries in the bigrams/trigrams lists were annotated as entities, while the rest are other types of collocations. Second, named entities are very prevalent at the very top of the list and scarce otherwise. This can be seen visually: both samples of 500 bigrams/trigrams are sorted by their rank according to our PMI_n measure (for bigrams it coincides with regular PMI), and the Entity / Not Entity tag is given beside it. Quantitatively, for bigrams, 16% of the 500 sampled bigrams are entities, but by refining the view into highest and lowest ranking PMI groups we get that 50% of the top 100 are entities while only 1% of the bottom 100 are entities. For trigrams 14% of the 500 sampled trigrams are entities, 37% of the top 100 are entities while only 4% of the bottom 100 are entities.  This breakdown perhaps can illuminate the natural intuition regarding PMI capturing entities -- indeed the top ranking PMI entries are largely entities. But we chose a much larger PMI-based masking vocabulary (see appendix 1 on the process of choosing its size), and the proportion of entities drops to around 1/7, with many of the added entries representing other collocations. We encourage the reviewers and readers to examine the other 6/7 entries that emerge after the very top of the list.
>
> (2) experimental analysis incorporating some entity-based masking into the experiments in the paper:
> In the camera ready versions we will include a comparison to entity masking based on NER, to be conducted on equal grounds within the framework in which we did all of the preceding experiments. This is to examine the effect of the 6/7 non-entity collocations on the performance. We expect there to be an effect, and expect our method to surpass entity masking (and in fact we expect random span masking to surpass entity masking as was found in Ref [2]), but there is no harm in reaffirming this in a controlled framework.
>
> Q3: We will make it clearer earlier in the paper that we consider only contiguous spans to be masked as units. The idea of jointly masking correlated non-neighboring tokens seems like a good one, but requires a different solution than the one proposed in this paper: the number of these blows up combinatorially with corpus size and n, whereas the number of n-grams is linear in both. We will include this in our discussion of future work (see more in our response to R2).
>
> Q4: That is a good point. Since PMI-Masking performed well, and since our focus was on demonstrating the effectiveness of the method, we did not attempt to "mix-in" some random token masking. We did consider it in our initial plans in order to mitigate the risk that PMI-Masking might pose prediction tasks that were too difficult for the MLM; however, when this did not occur, we shelved the idea. Still, there may be some value in it, both (1) in terms of speeding up the training at the beginning of pretraining; and (2) related to a point raised by R3, allowing the model to strengthen its lexical capabilities (represented by high perplexity as well as performance on targeted diagnostics) and thus enjoy the best of all worlds. We will include ablation on this and its effect on QA benchmarks/GLUE tasks as well as on perplexity/GLUE diagnostics in the camera ready version.
>
> [1] “ERNIE: Enhanced representation through knowledge integration”, Sun et al. (2019)
>
> [2] “Spanbert: Improving pre-training by representing and predicting spans”, Joshi et al. (2020)

---

### Official Review · AnonReviewer4 · 2020-10-29
**Solid theoretical and practical contribution, with experiments to back it up**

**Rating:** 7
**Confidence:** 4

**Review:**

Summary: This paper presents a masking strategy for training masked language models (MLM). The proposed strategy builds on previous approaches that mask semantically coherent spans of tokens (such as entire words, named entities, or spans) rather than randomly masking individual tokens. Specifically, the proposed method computes the PMI of spans (and the generalization for spans of size >2) over the pretraining corpus, and randomly masks from among the 800K spans (lengths 2-5) with the highest PMI. Masking based on PMI removes the ability for the model to rely on highly local signals to fill in the mask and instead focus on learning higher level semantics. They motivate this hypothesis with an experiment demonstrating that as the size of the WordPiece vocabulary decreases (and words are more frequently split into multiple tokens rather than being their own token), the transfer performance of the resulting MLM decreases. However, using whole-word masking with this same vocabulary size recovers much of the original performance, indicating that allowing the model to rely on these strong local signals harms the transfer quality of the resulting model.

Experiments: The paper evaluates on three standard NLU benchmarks: SQuAD2, GLUE, and RACE. They compare their PMI-based algorithm against random token masking, random span masking, and a naive version of their PMI masking strategy. All baselines are implemented within a single codebase. Their strategy outperforms random-span masking on SQuAD throughout pretraining, though the latter does close the gap on a smaller pretraining corpus (16GB). They show this gap remains if they use a larger pretraining corpus (54GB), as would likely be the case with a large-scale pretraining experiment. The resulting models also consistently outperform all other baselines on all tasks. They do additionally compare to outside models (AMBERT, SpanBERT, RoBERTa) and show their PMI-based models outperform or perform similarly to these models.
Overall, this paper introduces a solid theoretical basis for existing and new methods for training masked language models. They present robust experiments demonstrating the efficacy of their method under various settings.


1. Missing citation for RACE and original SQuAD dataset?
2. I assume this doesn’t happen often, but is it ever the case that a training example does not have any mask-able spans?

---

> ### Author Response · Authors · 2020-11-19
> **Thank you! Almost all training examples have mask-able spans**
>
> Thank you for your supportive feedback. We will include missing citations. Re not having any mask-able spans, on average around 50% of corpus tokens are identified as forming part of an n-gram on our list. So though in theory it is possible to imagine a sentence without mask-able spans (in which case our method reduces to the baseline whole-word masking), such sentences would occur very rarely.

---

### Official Review · AnonReviewer3 · 2020-10-29
**Clear empirical gains and well written, but somewhat incremental contribution overall**

**Rating:** 6
**Confidence:** 4

**Review:**

Summary:

The paper proposes a variant on the MLM training objective which uses PMI in order to determine which spans to mask. The idea is related to recently-proposed Whole Word Masking and Entity-based masking, but the authors argue the PMI-based approach is more principled. The method is straightforward--it involves computing PMIs for ngrams (in this case, up to length 5) over the training corpus, and then preferring to mask entire collocational phrases rather than single words during training. The intuition is that masking single words allows models to exploit simple collocations, thus optimizing their training objective without learning longer-range dependencies or higher level semantic features of the sentences, and this makes training less efficient than it could be. One contribution of the paper is a variant on the PMI metric that performs better for longer phrases by reducing the scores of phrases that happen to contain high-PMI subphrases, e.g. "George Washington is" should not have a high score despite the fact that "George Washington" does have a high score.

The authors compare their method against vanilla BERT with random masking, as well as against recently proposed variants such as SpanBERT and AMBERT, and show consistent improvements in terms of final performance as well as better efficiency during training. By way of analysis, the authors also make an argument that token-level perplexity is not correlated with downstream performance. This is an interesting point to make, though they do not expound upon it in this paper.

Strengths:

* The proposed method is simple and principled
* The empirical results show consistent improvement on standard benchmark tasks
* The proposed variation the PMI metric is a nice sub-contribution

Weaknesses:

* A somewhat marginal contribution, its not significantly different from the variants proposed previously (e.g., SpanBERT, entity masking)
* The evaluation focuses purely on benchmark tasks which are known to have flaws (e.g., the current "superhuman" performance on these tasks already makes gains on them suspect). I'd have liked to some more analysis/discussion of the linguistic consequences of this new objective. See more specific comments below.

Additional Comments/Questions:

I am curious about the more general effect of this training objective on the models linguistic (and particularly syntactic) knowledge. E.g., can you say more about how often the model sees unigrams being masked and how the distribution of these unigrams differs from what would be seen if we did random masking? I ask because I could imagine that this objective has a noticeable effect on the masking of function words (e.g., preposition occurring more often in collocations, pronouns and determines maybe less often) and thus the model might get differing access to these words in isolation. Since function words carry a lot more signal about syntactic structure than do content words and phrases (of the type you are capturing in your PMI metric), I'm very curious if there are some tradeoffs (or, possibly, additional advantages) that comes with your method that are not reflected by the benchmark performance metrics. Squad and GLUE are going to favor knowledge of things like entities and events, and capture very little about more nuanced linguistic reasoning, so reporting performance on some more recently released challenge sets, or using some probing studies, or at least just giving some analysis of win/loss patterns, would be very informative for assessing the contribution of this paper to NLP more generally.

---

> ### Author Response · Authors · 2020-11-19
> **Thank you!**
>
> Thank you for your thoughtful review, and both your positive and constructive comments. Due to the length of our reply we divide it to three comments as follows.

---

> > ### Author Response · Authors · 2020-11-19
> > **Regarding degree of similarity to previous approaches (SpanBERT, entity masking)**
> >
> > We generally see our work as unifying the different intuitions behind both approaches (along with whole-word masking) under a single premise, which makes it possible to improve MLM training in a principled manner.  Importantly, as we discuss in our response to R2, this unifying idea also suggests further ways that masking strategies might be improved in both text and in other domains.
> >
> > Beyond unifying earlier intuitions on how to improve random-token masking, our approach results in a qualitatively different masking scheme.  Unlike entity masking (and like span masking), our approach (1) is not dependent on the performance of NERs/parsers (which renders it applicable to any corpus, even in domains and languages for which parsing performance is poor) and (2) is not focused on a specific type of correlated span. There are many named entities in our PMI list, but they constitute a small percentage of the list overall (around 1 in 7) -- see more on this point in our response to R1's Q1&2. It seems that the intuition regarding PMI having a strong overlap with entities is correct for the very high ranking entries, but less correct as the list goes on. We attach randomly sampled bigrams and trigrams from the masking list, labeled as entity / not entity, in the supplementary material. Unlike random span masking, we focus only on spans that rank highly according to a principled statistical criterion, yielding a considerable improvement in bottom-line performance.

---

> > > ### Author Response · Authors · 2020-11-19
> > > **Regarding your question on unigram distribution**
> > >
> > > 50% of the masked spans in the PMI-Masking scheme were unigrams, as opposed to 100% in the random-token masking scheme and 20% in the random span masking scheme. We have uploaded to the supplementary material three charts containing the percentage of masked spans as a function of their length, one for each masking scheme. The average span length is 1.89 for PMI masking, and more than double that for random-span masking. This means that though we mask unigrams less than single-token masking, our approach resulted in significantly shorter average span lengths than the best setting of random span masking reported in Ref [1]. It seems that our focus on highly collocating spans achieved more effectiveness while covering shorter spans, allowing for more unigrams to be masked.
> > >
> > > Regarding the identity of these unigrams, we have uploaded to the supplementary material a table that contains two lists: the top 1000 word tokens sorted by the frequency in which *they were masked as unigrams* for Random-Token Masking and for PMI-Masking. Notably, perhaps addressing some of your concerns, there is an overlap of 85% between these two lists. Generally, the top 1000 frequent words contain many more function words than content words, so the top 1000 words masked in random token masking likely emphasize function words. The fact that this remains the case in PMI-Masking makes sense simply due to their prevalence in the corpus, and may also be reinforced by our modification to the naive PMI extension, which we expect to separate between a common function word and a neighboring high PMI span, so function words would still be masked as unigrams much of the time.
> > >
> > > To sum up, in PMI-Masking, unigrams are masked less than in random-token masking (around half as often), but the top frequency words masked as unigrams still include function words (resembling the top frequency words masked as unigrams in random-token masking), and these are very prevalent. Thus, we do not expect a 2x reduction to be dramatic.
> > > Having said all of this, our Table 4 does clearly shows that PMI-Masking yields a model with much higher single-token perplexity than that of  a model trained by Random-Token Masking or Random-Span Masking, which is as you note an interesting point to continue investigating. We conjecture that this is largely related to test/train discrepancy issues, as the input in the single-token perplexity task is less resemblant of training examples seen by the PMI-Masked model, but this may have further implications on, e.g., lexical semantic abilities of the model, to be discussed in the following.

---

> > > > ### Author Response · Authors · 2020-11-19
> > > > **Regarding evaluation on challenge sets**
> > > >
> > > > We thank you for this direction, we agree that it would reinforce the paper’s message. We will add a more nuanced discussion about what linguistic phenomena the model captures in the camera ready version. The detailed GLUE diagnostics scores of all the linguistic categories and sub-categories of the PMI Masking Model and the Random Span Masking baseline (received in the GLUE submission) are given in the supplementary. PMI-Masking surpasses the baseline in the overall score. Out of the 4 linguistic categories it is behind on Lexical Semantics and Logic, and ahead on Predicate Argument Structure and Knowledge (the latter by a large margin). This outcome is generally in line with the expectation from this masking scheme: knowledge will involve more phrases, entities, and concepts, and predicate argument structure can also benefit from a global understanding of the correct “segmentation” of the text. On the other hand, lexical semantics can be tied to understanding single tokens, and some of the logic tasks include negation, which may require sensitivity to single tokens.
> > > >
> > > > Please see our response to the 4th question of R1: we expect that mixing-in some Random-Token Masking will allow the model to strengthen its lexical capabilities (represented by high perplexity as well as these targeted diagnostics, while retaining its current advantages, enjoying the best of all worlds. We will include ablation on this and its effect on GLUE diagnostics/perplexity on the one hand, and QA-benchmarks/GLUE tasks on the other hand, in the camera ready version.

---

### Official Review · AnonReviewer2 · 2020-10-29
**Clearly written, simple idea, showed through extensive (well set-up) experiments to be very effective**

**Rating:** 8
**Confidence:** 4

**Review:**

This paper introduces an approach for masked language modeling, where they mask wordpieces together which have high PMI. The idea is relatively simple, has potential for high impact through broad adoption, and the paper is clearly written with extensive experiments.

The experiments on GLUE, SQuAD, and RACE are very well set up. For example, evaluating multiple learning rates for each downstream task is expensive, but really adds to confidence I have in the results. Reporting the best median dev score over five random initializations per hyperparameter, then evaluating that same model on the test set, definitely improves the reproducibility of the results. In addition, showing how performance is affected by the amount of pretraining data is very useful, and the experiments range from small scale to large scale. The ablations adjusting the vocabulary size (which, in turn, changes the size of wordpiece tokens) is a valuable contribution, and I would have asked to see something like this if it wasn't included. Table 4 is a nice addition -- it's interesting that the MLM loss is not predictive of downstream performance.

I suspect this approach will become widely adopted (or built upon) in future work pretraining language models. I give this paper an 8, only taking off points because the idea is relatively intuitive and doesn't really open a broad new area for future work. I don't see any obvious methodological flaws, which frankly, we can find in most papers.

I would be interested in seeing if this reduced the variance of the fine-tuning process. That might be something the authors could include for the camera ready, maybe in the appendix.

Edit: After reviewing the author response, I will keep my score as it is. I believe the paper should be accepted.

---

> ### Author Response · Authors · 2020-11-19
> **Thank you! Some discussion points**
>
> Thank you for encouraging feedback.
>
> As you recommend, we will add details on how the variance of the fine-tuning process is affected by the pretraining masking scheme. Your comment on our likely impact on future work got us thinking, which yielded two additional discussion points we plan to include in the camera ready version.
>
> The first is related to a point mentioned by R1 (in their third question). Our paper clearly demonstrates that by cutting off (masking) easy hints, it is possible to improve MLM pretraining. But we show this on the simplest-to-catch class of easy hints (which we show nonetheless to be effective): hints stemming from words that are immediately adjacent to the masked word to be predicted, forming contiguous spans. However, there are many non-local hints in language (e.g., R1's example), and we imagine that masking them could similarly allow the model to focus on deeper signals and further improve. Since looking for non-contiguous hints implies a much larger search space, there is no trivial way to traverse all of them. We therefore see extending the principles of our paper to non-contiguous hints as an important future direction.
>
> The second point relates to the potential impact of our work on self-supervision schemes employed in other fields, e.g., computer vision. We demonstrate that self-supervision can be improved by using the dependencies among input subsets for controlling the hardness and informativity of the self-supervision task. While we focused on the MLM self-supervision task in language, we note that researchers studying other domains are increasingly examining tasks involving recreating missing or distorted input segments, e.g. for creating self-supervised image representations [1]. Similarly to the original BERT, input patches for recreation are chosen at random. We believe that ideas from our paper could be used to motivate a more deliberate selection of the input patches to jointly recreate, thereby improving the representations attained by self-supervision.
>
> [1] “Selfie: Self-supervised pretraining for image embedding”, Trinh et al. (2019)

---

### Decision · Program_Chairs · 2021-01-07
**Final Decision**

**Decision:**

Accept (Spotlight)

**Comment:**

This paper describes a new and experimentally useful way to propose masked spans for MLM pretraining, by masking spans of text that co-occur more often than would be expected given their components - ie that are statistically likely to be non-compositional phrases.

The authors should make some attempt to connect their PMI heuristic with prior methods for statistical phrase-finding and term recognition, eg https://www.aaai.org/Papers/IJCAI/2007/IJCAI07-439.pdf or https://link.springer.com/chapter/10.1007/978-3-540-85287-2_24 in the final paper.